# *Morus alba* Root Extract Induces the Anagen Phase in the Human Hair Follicle Dermal Papilla Cells

**DOI:** 10.3390/pharmaceutics13081155

**Published:** 2021-07-27

**Authors:** Jiyu Hyun, Jisoo Im, Sung-Won Kim, Han Young Kim, Inwoo Seo, Suk Ho Bhang

**Affiliations:** 1School of Chemical Engineering, Sungkyunkwan University, Suwon 16419, Korea; guswldb97@skku.edu (J.H.); treeskb@skku.edu (J.I.); tjdnjdl90@skku.edu (S.-W.K.); tjdlsdn98@skku.edu (I.S.); 2Center for Theragnosis, Biomedical Research Institute, Korea Institute of Science and Technology, Seoul 02792, Korea; hy0408@kist.re.kr

**Keywords:** hair follicle dermal papilla cells, *Morus alba*, herbal extract, anagen-inducing, growth factor secretion

## Abstract

Restoring hair follicles by inducing the anagen phase is a promising approach to prevent hair loss. Hair follicle dermal papilla cells (HFDPCs) play a major role in hair growth via the telogen-to-anagen transition. The therapeutic effect of *Morus alba* activates β-catenin in HFDPCs, thereby inducing the anagen phase. The HFDPCs were treated with *M. alba* root extract (MARE) to promote hair growth. It contains chlorogenic acid and umbelliferone and is not cytotoxic to HFDPCs at a concentration of 20%. It was demonstrated that a small amount of MARE enhances growth factor secretion (related to the telogen-to-anagen transition). Activation of β-catenin was observed in MARE-treated HFDPCs, which is crucial for inducing the anagen phase. The effect of conditioned medium derived from MARE-treated HFDPCs on keratinocytes and endothelial cells was also investigated. The findings of this study demonstrate the potency of MARE in eliciting the telogen-to-anagen transition.

## 1. Introduction

Alopecia is a common disorder in modern society. Hair loss causes vary, including hormone-induced reasons, drug treatment, and nutrient deficiency [1,2]. Hair loss can be restored by hair follicle regeneration [3], which is affected by keratinocytes, fibroblasts, and dermal papilla cells [4,5].

Hair follicle dermal papilla cells (HFDPCs) are located at the base of hair follicles. They play pivotal roles in regulating the phases of hair growth: growth phase (anagen), regression/transition phase (catagen), and rest phase (telogen) [6]. Telogen-to-anagen phase transition is required for continuous hair growth, but the effects of physiological and emotional stress can block this transition and cause alopecia [7]. In each state and state transition, HFDPCs show different gene expression [8,9]. For the transition to anagen state, vascular endothelial growth factor (VEGF) and keratinocyte growth factor (KGF), also known as fibroblast growth factor 7 (FGF7), play a major role [10,11].

Rajendran et al. used extracellular vesicles derived from mesenchymal stem cells to promote VEGF and KGF upregulation in the HFDPCs and showed hair regrowth in C57BL/6 mice. Minoxidil (an FDA-approved, well-known chemical compound) also upregulates the expression of VEGF in HFDPCs to promote hair growth [12,13,14]. Activation of the Wnt/β-catenin pathway, which can prevent degradation of β-catenin and translocate to the nucleus to promote anagen-related gene expression, plays a major role in initiating the anagen phase [15,16]. Adverse side effects of minoxidil and Propecia (another FDA-approved medicine against hair loss) include allergic reactions, contact dermatitis, itching, dryness, scalp irritation [17,18], psychiatric symptoms, and adverse sexual effects [19]. Herbal extracts induce a telogen-to-anagen transition with low side effects. *Polygonum multiflorum* extract showed an increase in growth factors (IGFBP2, PDGF, and VEGF) in the HFDPCs. A similar effect was observed in HFDPCs treated with the *Hottuynia cordata* extract. The expression of PDGF-AA and VEGF is increased by the phosphorylation of extracellular signal-regulated kinase (ERK) and protein kinase B (AKT) [20,21].

The therapeutic effects of *Morus alba* root extract (MARE) include antidiabetic and antioxidant effects [22,23], as well as anti-cancer activities [24], and it also improves symptoms of osteoarthritis [25]. This extract contains chlorogenic acid and umbelliferone, which exert antioxidant effects [26,27,28] and promote hair growth [29,30,31,32].

Previous studies have only focused on the expression of several genes and proteins in HFDPCs. Therefore, the effect of growth factors secreted by HFDPCs on nearby cells, such as keratinocytes and endothelial cells, requires further study. Proliferation and differentiation of keratinocytes along with recruitment of blood vessels are crucial to activate the anagen state related to hair follicle formation [33,34]. This study analyzed the components of MARE and its concentration-dependent effect on the viability of HFDPCS. After optimizing MARE concentration, the increase in representative growth factors—KGF, VEGF, fibroblast growth factor 2 (FGF2), and hepatocyte growth factor (HGF), along with β-catenin activation, an important marker for telogen-to-anagen transition—were evaluated. Furthermore, this study investigated the effect of MARE-treated, HFDPC-derived conditioned medium (CM) on keratinocytes and endothelial cells (responsible for hair growth in HFDPCs).

## 2. Materials and Methods

### 2.1. Extraction of M. alba Root Extract and Liquid Chromatography–Tandem Mass Spectrometry Analysis

MARE was purchased from Caronbio (Seoul, Korea). Dried *M. alba* root (1 kg) was boiled in 10 L of distilled water for 15 h at 90 °C and then filtered using a 0.45 μm filter paper. The liquid chromatography–tandem mass spectrometry (LC–MS/MS) system included an Agilent 1200 liquid chromatography system equipped with a binary pump, vacuum degasser unit, autosampler, and Agilent 6470 B LC/TQ (Agilent, Santa Clara, CA, USA) with an electrospray ion (ESI) source. Chromatographic separation was achieved on a Poroshell 120 EC-C18 column (Agilent, Santa Clara, CA, USA) (3.0 × 50 mm, 2.7 µm) at 40 °C. The mobile phase consisted of 0.1% formic acid in water (A) and 0.1% formic acid in acetonitrile (B) using a gradient elution of 80(A):20(B) at 0–3 min, 20(A):80(B) at 3–3.1 min, and 80(A):20(B) at 3.16 min. The flow rate was set at 0.5 mL/min. The sample injection volume was 5 μL. Complete data acquisition and peak integration were performed using Analyst software (version 1.4.2) from Applied Biosystems/MDS Sciex (SCIEX, Redwood city, CA, USA). The analytes were determined in electrospray ionization mode and quantified using the multiple-reaction monitoring (MRM) mode. Mass spectrometry was performed with an optimized capillary voltage of 3500 V, gas temperature of 270 °C, flow rate of 7 l/min, and sheath gas temperature of 350 °C with a flow rate of 11 l/min flow. The nebulizer gas pressure was maintained at 40 psi.

### 2.2. Cell Culture

The HFDPCs were purchased from PromoCell (Heidelberg, Germany) and cultured in Follicle Dermal Papilla Cell Growth Medium Kit (PromoCell) supplemented with 1% (*v*/*v*) penicillin/streptomycin (PS, Gibco BRL). Normal human epithelial keratinocytes (NHEKs) were purchased from PromoCell and cultured in keratinocyte growth medium 2 (PromoCell) supplemented with 1% (*v*/*v*) PS. The cells were incubated in 150-mm cell culture Petri dishes (Corning Inc., Corning, New York, NY, USA) until they reached 70–80% confluence. The cells were incubated at 37 °C and 5% CO_2_ saturation. The cell culture medium was changed every 3 days. Cells within 4 passages were used for the experiments.

### 2.3. Cell Counting Kit-8 Assay

Cell proliferation was analyzed using the Cell Counting kit-8 (CCK-8; Sigma-Aldrich, St. Louis, Missouri, MO, USA). The HFDPCs were seeded in 24-well plates (2 × 10^4^ cells/well), and cell proliferation was measured with CCK-8 for 1 d after each percentage of MARE treatment. The cells were washed in phosphate-buffered saline (PBS) and replaced with culture medium containing CCK-8 solution (10%, *v*/*v*). After 3 h of incubation at 37 °C, the absorbance of each well was measured at 450 nm (Infinite F50, TECAN, Männedorf, Switzerland).

### 2.4. Live and Dead Assays

Live and dead assays were performed using fluorescein diacetate (FDA, Sigma-Aldrich) and ethidium bromide (EB, Sigma-Aldrich) staining. The cytoplasm of viable cells and nuclei of nonviable cells were stained with FDA (green) and EB (red), respectively. The staining solution was freshly prepared by combining 10 mL of FDA stock solution (1.5 mg/mL of FDA in dimethyl sulfoxide), 5 mL of EB stock solution (1 mg/mL of EB in PBS), and 3 mL of PBS. The HFDPCs were cultured for 24 h with MARE extract, washed once with PBS, and incubated with the staining solution for 3 min at 37 °C. After staining, the samples were washed twice with PBS and examined under a fluorescence microscope (DFC 3000 G, Leica, Wetzlar, Germany).

### 2.5. Cellular Morphology

The morphology of the HFDPCs was examined using an optical microscope (Olympus CKX53, Tokyo, Japan). The HFDPCs were seeded in 6-well plates at a density of 1 × 10^5^ cells/well, treated with each percentage of MARE extract the next day, and cultured for the next 24 h. Thereafter, the cells were fixed with 4% paraformaldehyde in PBS for 10 min at 25 °C and examined under a microscope.

### 2.6. Apoptotic Activity

The HFDPCs were seeded in 6-well plates at a density of 1 × 10^5^ cells/well and treated with each percentage of MARE extract after 24 h. The HFDPCs were then cultured for the next 24 h and fixed with 4% paraformaldehyde in PBS for 10 min at 25 °C. A terminal deoxynucleotide transferase-mediated deoxyuridine triphosphate nick-end labeling (TUNEL) assay was performed using the ApopTag^®^ Fluorescent In Situ Apoptosis Detection Kit (Millipore, Bedford, MA, USA) to examine the apoptotic activity of the HFDPCs. After 4,6-diamidino-2-phenylindole (DAPI, Vector Laboratories, Burlingame, CA, USA) staining, TUNEL-positive fluorescence was measured using a fluorescence microscope (IX71 inverted microscope, Olympus).

### 2.7. Phalloidin Staining

The HFDPCs were seeded in 6-well plates at a density of 1 × 10^5^ cells/well and treated with each percentage of MARE extract after 24 h. HFDPCs were then cultured for the next 24 h and subsequently fixed with 4% paraformaldehyde in PBS for 10 min at 25 °C. Thereafter, phalloidin L (Vector Laboratories, Burlingame, CA, USA) was added to the samples for 10 min. After staining, the samples were washed twice with PBS and counterstained with DAPI. The cells were then examined under a fluorescence microscope (DFC 3000 G, Leica, Wetzlar, Germany).

### 2.8. Quantitative Reverse Transcription Polymerase Chain Reaction

Quantitative reverse transcription polymerase chain reaction (qRT-PCR) was used to quantify the relative gene expression levels of human *VEGF, FGF7, HGF, FGF2,* and *β-catenin* among the experimental groups. Human-specific gene primers were used for in vitro HFDPC and NHEK analyses. Total ribonucleic acid (RNA) was extracted from the samples using 1 mL TRIzol reagent (Life Technologies, Inc., Carlsbad, CA, USA) and 200 μL chloroform. The lysed samples were centrifuged at 12,000 rpm for 10 min at 4 °C. The RNA pellet was washed with 75% (*v*/*v*) ethanol in water and dried. The samples were then dissolved in RNase-free water. The SsoAdvanceed Universal SYBR Green Supermix kit (Bio-Rad, Hercules, CA, USA) and CFX Connect™ real-time PCR detection system (Bio-Rad) were used for qRT-PCR. Table 1 lists the primers used for qRT-PCR.

### 2.9. Western Blotting

The HFDPCs were seeded in 100 mm cell culture Petri dishes (5 × 10^5^ cells) and treated with each percentage of MARE extract after 24 h. The HFDPCs were then cultured for the next 24 h and lysed in RIPA buffer (Rockland Immunochemicals Inc., Limerick, PA, USA). After centrifugation at 10,000× *g* for 10 min, the supernatant was prepared as a protein extract. Protein concentrations were determined using a BCA assay (Pierce Biotechnology, Rockford, IL, USA). Equal amounts of protein from each sample were mixed with a sample buffer and subjected to sodium dodecyl sulfate polyacrylamide gel electrophoresis (SDS-PAGE) using a 10% (*v*/*v*) resolving gel. The separated proteins were transferred to immune-blot PVDF membranes (Bio-Rad). The membranes were blocked with 5% (*w*/*v*) skim milk in Tris-buffered saline (TBS; 50 mM Tris–HCl (pH 7.5), 150 mM NaCl, 2.5 mM KCl) and incubated for 1 h at 25 °C. Then, the membranes were probed overnight at 4 °C with antibodies against glyceraldehyde 3-phosphate dehydrogenase (GAPDH) (Abcam, ab9485, 1:2000, Cambridge, UK), VEGF (Abcam, ab46154, 1:1000), FGF7 (Abcam, an131162, 1:1000), and β-catenin (Abcam, ab223075, 1:1000). Thereafter, the membranes were incubated with horseradish peroxidase-conjugated secondary antibody (R&D Systems, HAF008 for GAPDH, 1:2000 HAF017 for VEGF, FGF7, and β-catenin, 1:1000, Minneapolis, MN, USA) for 1 h at 25 °C, followed by addition of ECL reagent (TransLab, Daejeon, Korea). The blots were developed in a dark room, and luminescence was recorded using an X-ray blue film (Agfa HealthCare NV, Mortsel, Belgium).

### 2.10. Immunocytochemistry

The HFDPCs were seeded in 6-well plates at a density of 1 × 10^5^ cells/well and treated with each percentage of MARE extract after 24 h. HFDPCs were then cultured for the next 24 h and then fixed with 4% paraformaldehyde in PBS for 10 min at 25 °C. The cells were stained using the immunofluorescence method with anti-β-catenin antibodies (Abcam, ab223075, 1:100). The β-catenin-positive signals were visualized with fluorescein isothiocyanate-conjugated secondary antibodies (Jackson ImmunoResearch Laboratories, West Grove, PA, USA). Then, the cells were counterstained with DAPI and phalloidin, and the signals were examined under a fluorescence microscope (IX71 inverted microscope, Olympus).

### 2.11. Conditioned Medium Preparation

The HFDPCs were seeded in 6-well plates at a density of 1 × 10^5^ cells/well and treated with each percentage of MARE extract after 24 h. The HFDPCs were then cultured for the next 24 h, followed by medium replacement with HFDPC culture medium, and incubated for 72 h at 37 °C. The CM was centrifuged at 1500 rpm for 5 min to eliminate cell debris.

### 2.12. Tubular Formation Assay

Endothelial cell tube formation was assessed using an angiogenesis assay kit (ab204726, Abcam), according to the manufacturer’s instructions. Human umbilical vein endothelial cells (HUVECs) were seeded onto an extracellular matrix gel (2 × 10^4^ cells/well) in 100 μL of HFDPC-derived CM and incubated for 4 h at 37 °C. Following incubation, the HUVECs were stained with staining dye for 30 min at 37 °C and observed under a fluorescence microscope (IX71, Olympus).

### 2.13. Effect of Conditioned Medium Derived from Hair Follicle Dermal Papilla Cells on Proliferation and Involucrin Expression in Normal Human Epithelial Keratinocytes

NHEKs were seeded on 24-well plates (2 × 10^4^ cells/well) for the CCK-8 assay and on 6-well plates (1 × 10^5^ cells/well) for immunocytochemistry. After 24 h, the cells were treated with NHEK culture media and CM derived from each percentage of MARE extract-treated HFDPCs (1:1, *v/v*). The NHEKs were then cultured for 72 h, and the CCK-8 assay for NHEK proliferation and immunocytochemistry for involucrin (Abcam, ab53112, 1:500) was performed. Involucrin-positive fluorescence signals were visualized using fluorescein isothiocyanate-conjugated secondary antibodies (Jackson ImmunoResearch Laboratories). The cells were then counterstained with DAPI and phalloidin and examined under a fluorescence microscope (IX71 inverted microscope; Olympus).

### 2.14. Statistical Analyses

All data are presented as the mean ± standard deviation. Data were statistically analyzed using GraphPad Prism (GraphPad Software, San Diego, CA, USA). One-way analysis of variance (ANOVA) was used to determine statistical significance. Differences of *p* < 0.05 and *p* < 0.001 were considered significantly different from the control.

## 3. Results

### 3.1. Liquid Chromatography–Tandeom Mass Spectrometry Profiling of Chlorogenic Acid and Umbelliferone in M. alba Root Extract

Chlorogenic acid and umbelliferone in MARE were identified by LC–MS/MS analysis. Following the electrospray ionization mode for chlorogenic acid and umbelliferone, they were quantified in the MRM mode, and the transition of the precursor ion (chlorogenic acid: 355.1 Da, umbelliferone: 163.0 Da) to the product ion (chlorogenic acid: 89.1 Da, umbelliferone: 77.1 Da) was observed. For the calibration curve of analytes, a typical regression equation was applied to the calibration curve of the analytes (Table 2). For both chlorogenic acid and umbelliferone, the correlation coefficient was more than 0.99. High-performance liquid chromatography (HPLC) data showed that the retention times of chlorogenic acid and umbelliferone were 0.5 min and 1.5 min, respectively, which were identical to the standard for each compound (Figure 1A,B). The HPLC area of each sample was 8117 for chlorogenic acid (dilution: 200 mL/g) and 2387 (dilution: 2 mL/g) for umbelliferone. The concentrations of chlorogenic acid and umbelliferone were 36.05 ng/mL and 0.1 ng/mL, respectively. The mass spectrometry data performed in the MRM mode also demonstrated that MARE contained chlorogenic acid and umbelliferone (Figure 1C,D).

### 3.2. Optimizing the Concentration of M. alba Root Extract via the Viability of Hair Follicle Dermal Papilla Cells

The optimal concentration of MARE was determined by evaluating the viability of HFDPCs at different concentrations of MARE (10, 20, 30, 40, and 80 *v*/*v*, %). The viability of the HFDPCs was similar in the 0% group, 10% group showed 98% of live cells, and it was slightly decreased in the 20% group (80% compared to that of the 0% group). The viability of the HFDPCs in other groups dramatically decreased by 17%, 11%, and 11% in 30%, 40%, and 80% groups, respectively, compared to those of 0% group (Figure 2A, upper panel). Cell viability was analyzed using an FDA/EB assay. Few dead cells were observed in the 0–20% group, whereas dead cells were dominant in the other groups (Figure 2A: lower panel and Figure 2B). In terms of HFDPCs density after MARE treatment, similar density in unit area (mm^2^) was observed in the 0–20% group (103% in 10% group and 90% in 20% group compared to that of the 0% group). However, the 30–80% group showed decreased cell density (67% in the 30% group, 54% in the 40% group, and 32% in the 80% group compared to that of the 0% group), which was similar to the result observed in viability test (Figure 2C, left panel). In comparison with the HFDPCs showing normal cell morphology, in the 0–20% group, morphological changes in the MARE-treated HFDPCs showed similar cell viability. The HFDPCs in the 30–80% group showed diminished and shrunken cell morphology, indicating cell death (Figure 2C, right panel, circled) [35]. These results indicate that treatment with 20% MARE might be suitable for HFDPCs.

### 3.3. Cell Viability and Morphology Data Depend on M. alba Root Extract Concentration

Further assays were performed on the 0–20% group, which demonstrated that the non-cytotoxic effect of MARE under 20% concentration did not affect the viability of the HFDPCs. Using the TUNEL assay, we determined the number of apoptotic cells after MARE treatment. As shown by the arrows in Figure 3A, there was no significant difference in the number of apoptotic cells between the groups. Using phalloidin for F-actin staining, we found that the morphology of the HFDPCs was altered in accordance with the concentration of MARE, and their nuclei were stained with DAPI. The morphology of the HFDPCs in the 0–20% group was similar (Figure 3B). F-actin expression was slightly lower in the 20% group than in the 0% group, but there was no significant difference between the groups (Figure 3C). Thus, less than 20% of the MARE-treated HFDPCs did not show cytotoxicity.

### 3.4. Growth Factor Expressions in M. alba Root Extract-Treated Hair Follicle Dermal Papilla Cells

Angiogenesis-related genes (*VEGF, HGF, FGF2*) [36] and *FGF7* play a key role in stimulating the telogen-to-anagen transition [10,37]. To determine whether MARE treatment would improve growth factor secretion, we treated HFDPCs with various concentrations of MARE, and their gene and protein expression was compared to that of normal HFDPCs. The VEGF expression observed in the control group increased 3.90-fold (*p* < 0.05) and 5.84-fold (*p* < 0.001) in the 10% and 20% groups, respectively (Figure 4A). FGF7 expression was increased in both the 10% and 20% MARE-treated groups. Compared to the control (0%) group, a 1.63-fold (*p* < 0.001) increase in the 10% group and 2.07-fold (*p* < 0.001) increment in the 20% group (Figure 4B) was observed. Similar to FGF7 and VEGF, HGF and FGF2 expression also increased in the 10% (2.32-fold in *HGF*, 1.31-fold in *FGF2*) and 20% (2.95-fold in *HGF* and 2.02-fold in *FGF2*) groups, respectively, when compared to the control group (Figure A1). In each group, the growth factor protein expression of HFDPC was compared using Western blotting. The results showed that with an increase in MARE concentration, VEGF and FGF7 protein levels were upregulated (Figure 4E). Quantification of protein bands indicated that when comparing the 10% and 20% groups with the 0% group, the VEGF expressions were 5.16-fold and 10.21-fold higher and the FGF7 expressions were 1.36- and 1.72-fold higher, respectively (Figure 4D,E and Figure A2). These results indicate that MARE affects HFDPCs to increase the expression of angiogenesis-related factors (VEGF, HGF, FGF2, and FGF7), which are critical in the telogen-to-anagen transition and hair growth [11,33,38].

### 3.5. β-Catenin Expression in M. alba Root Extract-Treated Hair Follicle Dermal Papilla Cells

Activation of β-catenin in dermal papilla cells is crucial for inducing anagen state, growth factor secretion, and hair follicle formation [39,40,41]. To investigate β-catenin activation in HFDPCs using MARE treatment, we examined *β-catenin* gene expression using qRT-PCR in 0–20% MARE-treated HFDPCs. Compared to the 0% group, *β-catenin* expression was upregulated 1.59- (*p* < 0.078) and 1.97-fold (*p* < 0.05) in the 10% and 20% groups, respectively (Figure 5A). Protein expression was assessed using Western blotting and showed a similar trend as the gene expression; the bands increased with an increase in MARE concentration (Figure 5B). β-Catenin translocates into the nucleus to promote growth factor expression, thereby increasing its secretion [42,43]. Immunocytochemistry assay revealed that high-concentration MARE treatment increased β-catenin expression in the nucleus, which further increased the expression of growth factors (Figure 5C).

### 3.6. Effect of Conditioned Medium Derived from M. alba Root Extract-Treated Hair Follicle Dermal Papilla Cells on Human Umbilical Vein Endothelial Cells and Normal Human Epithelial Keratinocytes

Dermal papilla cells in human hair follicles are surrounded by keratinocytes and blood vessels with which they interact [44,45]. To analyze the effect of MARE-treated HFDPCs, we examined the effect of CM derived from HFDPCs in HUVECs and NHEK. In the tubular formation assay, relatively numerous and rapid tube formation was observed in HUVECs treated with high concentrations of MARE (Figure 6A,B), whereas in NHEKs, involucrin expression was relatively high in CM from the MARE-treated group (Figure 6C,D). Upregulated expression of involucrin is found in the maturation and development of anagen via keratinocyte maturation [34,46]. The proliferation of NHEK also increased with increasing MARE concentration (Figure 6E). These results suggested that MARE-treated HFDPCs secreted angiogenic related growth factors, as seen in HUVECs, and led to keratinocyte differentiation, a marker of anagen state.

## 4. Discussion

Despite its importance, the effect of MARE-treated dermal papilla cells on nearby cells such as keratinocyte or endothelial cells has rarely been investigated. Our study suggests that MARE has great potential in upregulation of angiogenic paracrine factor secretions from HFDPCs, which is crucial in telogen-to-anagen transition. Additionally, the paracrine factors secreted from MARE-treated HFDPCs promoted the angiogenic capacity of endothelial cells along with the proliferation and maturation of keratinocyte. HFDPCs and keratinocytes comprise the hair follicle structure and interact with each other [47]. During hair growth, each cell type involved in the transition from the telogen-to-anagen phase shows unique characteristics. HFDPCs play critical roles in regulating hair growth by managing hair follicle cycling through growth factor secretion [48]. HFDPCs activate β-catenin during telogen-to-anagen transition [49]. Proliferation and differentiation of keratinocytes, present in the hair follicle epithelium, increase during the anagen phase [34,46,50,51]. Angiogenesis is required to supply nutrients for hair growth via blood vessels [33,52].

MARE contains chlorogenic acid and umbelliferone. Chlorogenic acid is known for its antioxidant activity and the prevention of β-catenin phosphorylation, which induces β-catenin degradation. Thus, the β-catenin signaling pathway was activated [53]. It was also found that MARE containing high concentrations of chlorogenic acid activated β-catenin. The viability of HFDPCs treated with various concentrations of MARE, especially under 20% concentration, was not critically affected, but over 20% viability significantly decreased. Regardless of the MARE treatment, cell density and the cellular morphology of the HFDPCs did not change until the concentration of MARE reached 20%, beyond which the HFDPCs showed apoptotic bodies.

As the concentration of MARE increased, the mRNA and protein expression of several growth factors (KGF, VEGF, FGF2, and HGF) in the HFDPCs was significantly upregulated. FGF7, a member of the FGF family, is an important factor that leads to hair follicle proliferation and differentiation [54,55]. Well-known angiogenesis-related growth factors, VEGF, FGF2, and HGF, can stimulate hair follicle proliferation [11,33,38,56,57,58]. At both the gene and protein levels, VEGF levels increased significantly. An increase in pro-angiogenic factor expression may promote anagen transition and hair follicle growth. In terms of the mRNA and protein expression results, we found that MARE-treated HFDPCs showed significant enhancement of β-catenin expression with translocation of β-catenin from the cytosol to the nucleus. Activation of β-catenin is a result of the state of canonical Wnt signaling. After β-catenin activation, it moves into the nucleus to induce the transcription of growth factor [59]. In concordance with previous reports, increased expression of both β-catenin and other growth factors was observed. This result indicates the state of the Wnt/β-catenin pathway [60]. In conclusion, the upregulation of various growth factor secretions was induced by β-catenin activation (Figure 7), and MARE-treated HFDPCs restored the anagen state for hair growth. Keratinocyte proliferation and involucrin expression are critical to the anagen state along with the angiogenic microenvironment around the HFDPCs [34,61]. As shown in the results, the secretion of angiogenic paracrine factors from HFDPCs also increased with MARE treatment. CM derived from MARE-treated HFDPCs induced earlier tube formation in HUVECs, which was led by angiogenic growth factors in the CM [62]. As the concentration of MARE in the HFDPCs increased, increased levels of angiogenic growth factors were observed. In addition, the keratinocytes located near the HFDPCs showed improvement in their proliferation and expression of involucrin, depending on the MARE treatment concentration of the HFDPCs. Along with the enhanced proliferation of keratinocytes, an increase in involucrin expression was shown using immunocytochemistry (ICC) of CM-treated keratinocytes, which suggested another proof of anagen inducing potential [50,51,63]. These results support that MARE can initiate molecular mechanisms related to anagen inducement in HFDPCs affecting various cell types supporting hair growth.

## 5. Conclusions

In summary, the MARE extract contained high concentrations of chlorogenic acid and umbelliferone. Treating the HFDPCs with an optimized concentration of MARE did not affect the viability or morphological changes. The expression of growth factors that could induce anagen transition was increased in MARE-treated HFDPCs because of the activation of β-catenin. Results using CM derived from MARE-treated HFDPCs showed improved angiogenesis, which is critical for anagen transition. Enhanced proliferation and differentiation of keratinocytes revealed that MARE facilitated the telogen-to-anagen transition. This study elucidated the molecular mechanisms underlying the telogen-to-anagen transition induced by MARE. The keratinocytes and endothelial cells located near the HFDPCs were analyzed, as they play important roles in hair growth promotion. This study suggests that MARE has great potential for promoting hair growth by inducing telogen-to-anagen transition in terms of enhanced angiogenic paracrine factor secretion from HFDPCs and following effect on enhanced tubular formation in endothelial cells and proliferation and maturation in keratinocytes.

## Figures and Tables

**Figure 1 pharmaceutics-13-01155-f001:**
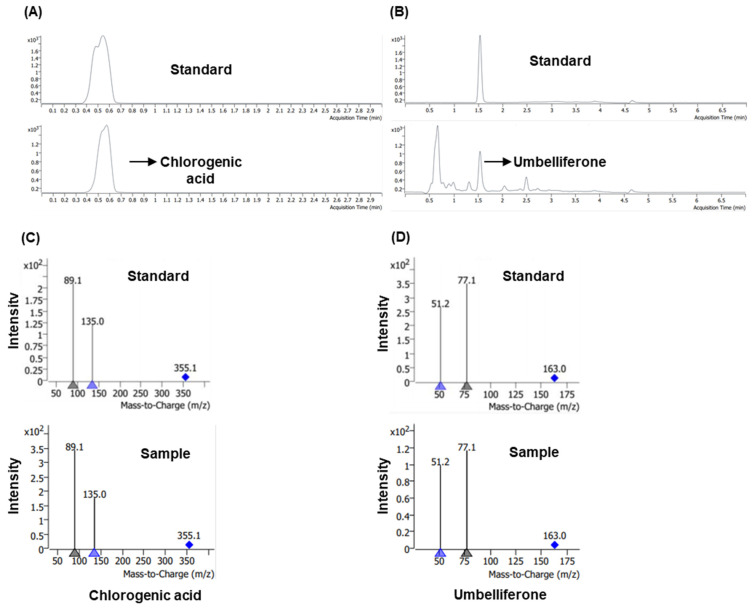
Concentration of chlorogenic acid and umbelliferone in *Morus alba* root extract (MARE). High-performance liquid chromatography (HPLC) chromatogram of (**A**) standard sample of chlorogenic acid and MARE (**B**) standard sample of umbelliferone and MARE. Product ion spectra of (**C**) chlorogenic acid standard (above) and sample (below) and (**D**) umbelliferone standard (above) and sample (below).

**Figure 2 pharmaceutics-13-01155-f002:**
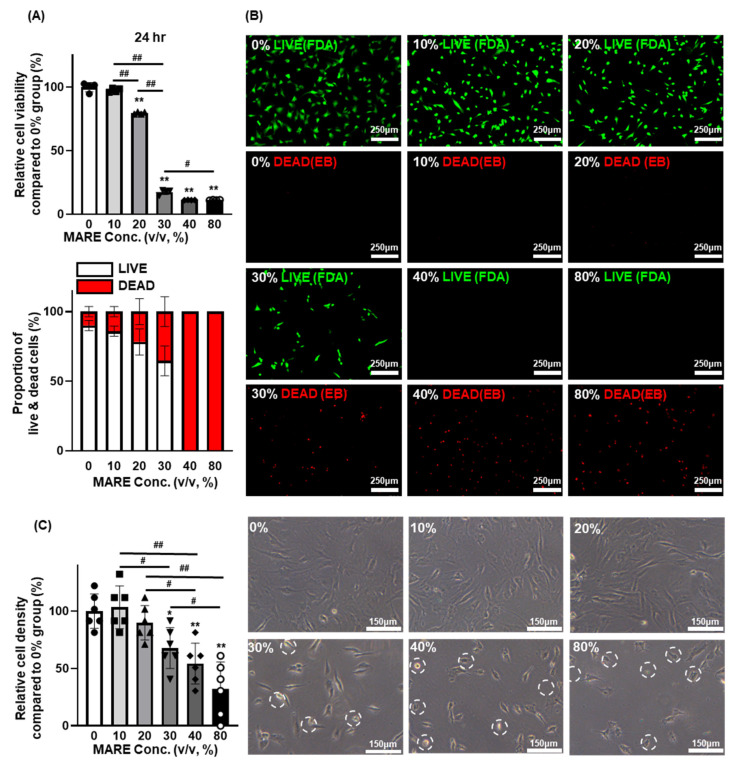
Optimized MARE concentration for hair follicle dermal papilla cells (HFDPCs) in terms of cell viability. (**A**) Cell counting kit-8 (CCK-8) assay results of MARE-treated HFDPCs (24 h, *n* = 4, upper panel) and quantification of EB-positive cells in unit area compared to the 0% group (*n* = 6, lower panel). * *p* < 0.05, ** *p* < 0.001 compared to the 0% group, # *p* < 0.05, ## *p* < 0.001 compared to each other. (**B**) Live (fluorescein diacetate (FDA), green) and dead (ethidium bromide (EB), red) assay results of MARE-treated HFDPCs. (**C**) Relative cell density of HFDPCs (*n* = 6, left panel) and light microscopy images showing cell morphology (right panel) of the HFDPCs after MARE treatment with various concentrations (100×). * *p* < 0.05, ** *p* < 0.001 compared to the 0% group, # *p* < 0.05, ## *p* < 0.001 compared to each other.

**Figure 3 pharmaceutics-13-01155-f003:**
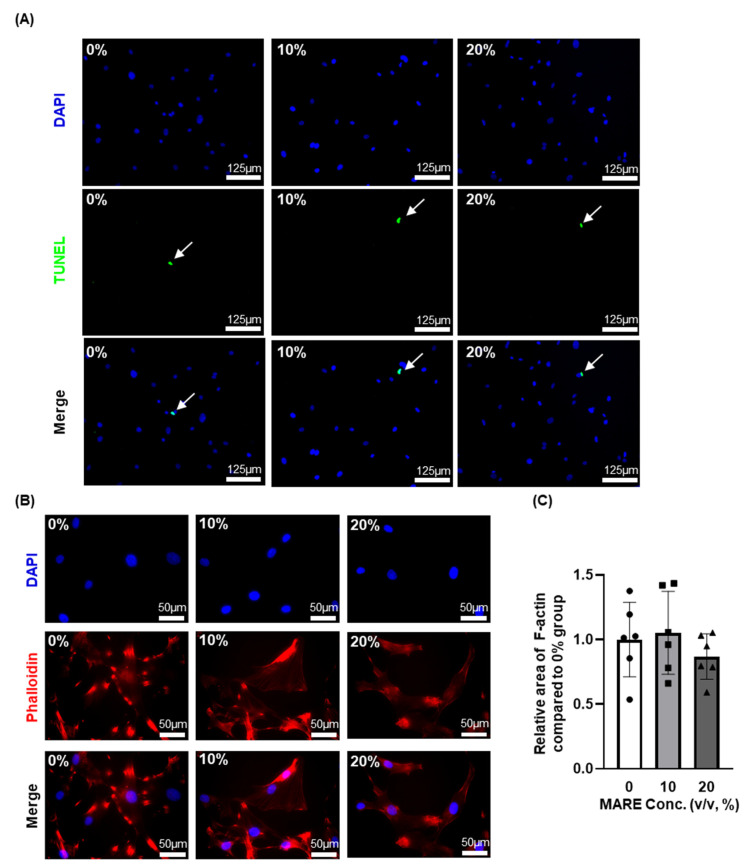
Effect of 0 to 20% concentration of MARE on the HFDPCs. (**A**) Terminal deoxynucleotide transferase-mediated deoxyuridine triphosphate nick-end labeling (TUNEL) assay results of MARE-treated HFDPCs. Cell nucleus was stained with 4′,6-diamidino-2-phenylindole (DAPI) (blue), and apoptotic cell was stained with fluorescent-dUTP (green, arrow marked). (**B**) DAPI and phalloidin staining results of MARE-treated HFDPCs showing F-actin expression. (**C**) Quantification of F-actin area compared to the 0% group (*n* = 6).

**Figure 4 pharmaceutics-13-01155-f004:**
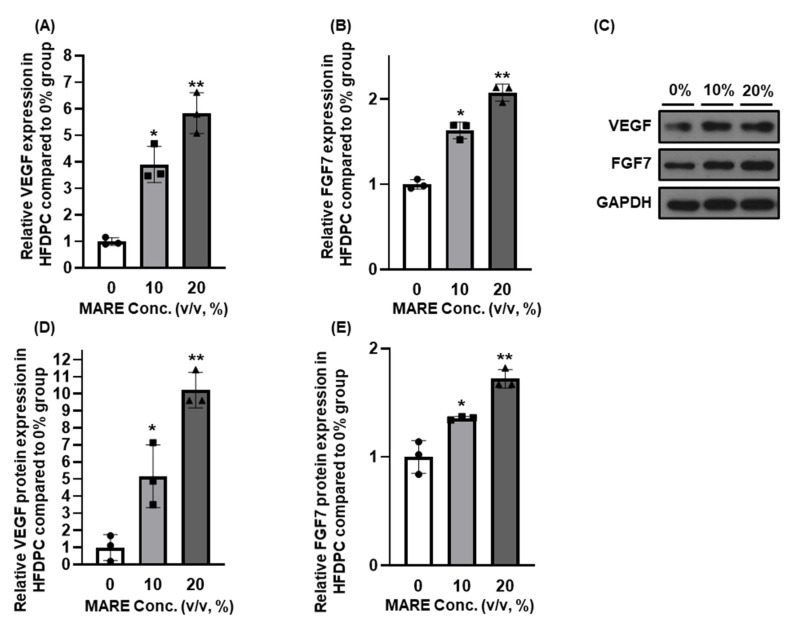
Effect of MARE on (**A**) vascular endothelial growth factor (VEGF), (**B**) fibroblast growth factor 7 (FGF7) (*n* = 3, * *p* < 0.05, ** *p* < 0.001 compared to the 0% group). Representative results of (**C**) VEGF, FGF7 protein expression, and their (**D**,**E**) quantification (*n* = 3, * *p* < 0.05, ** *p* < 0.001 compared to the 0% group).

**Figure 5 pharmaceutics-13-01155-f005:**
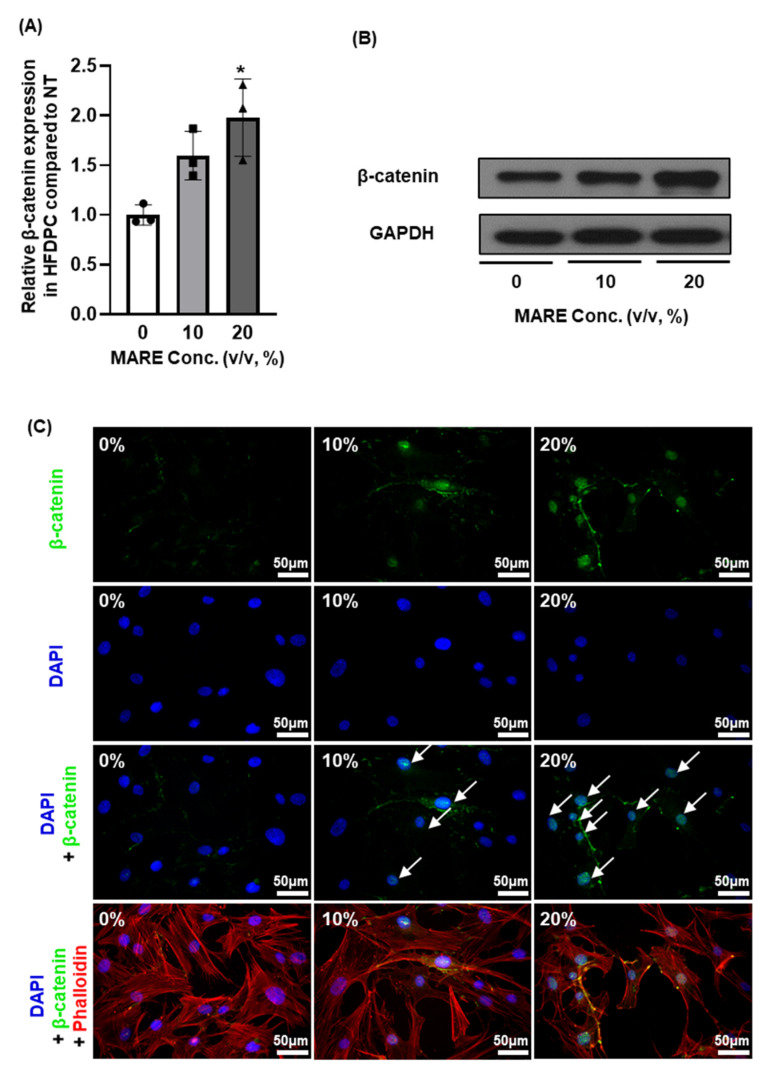
Effect of MARE on β-catenin activation in the HFDPCs. (**A**) Relative β-catenin expression in HFDPC assessed using qRT-PCR (*n* = 3, * *p* < 0.05). (**B**) Representative β-catenin protein expression in the HFDPC assessed using Western blotting. (**C**) Immunocytochemistry (ICC) of β-catenin (green) staining merged with DAPI (nucleus, blue) and phalloidin (F-actin, red) signals.

**Figure 6 pharmaceutics-13-01155-f006:**
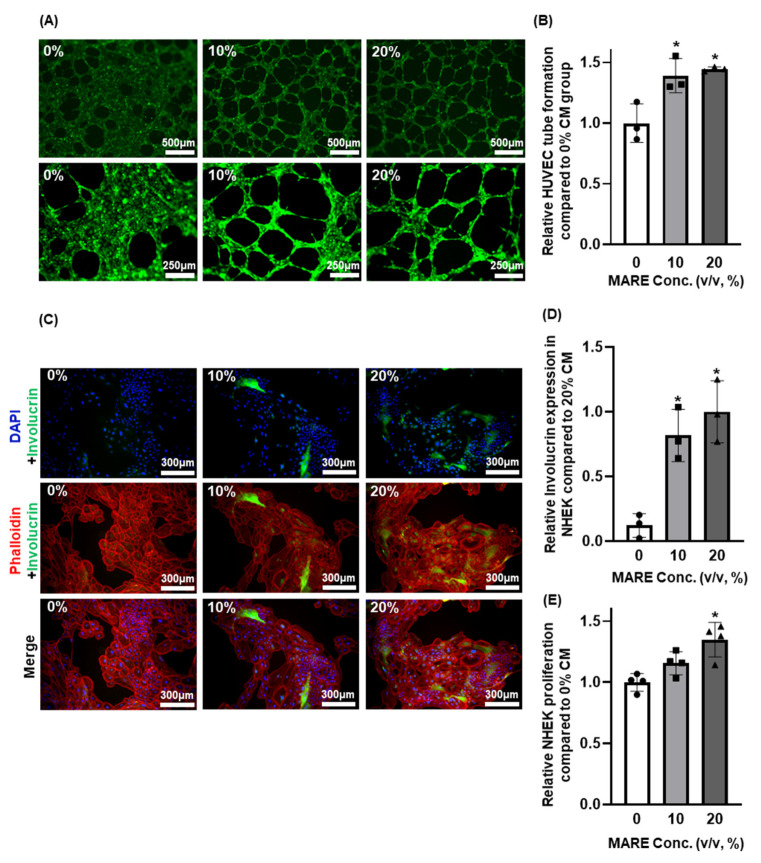
Effect of conditioned media (CM) retrieved from MARE-treated HFDPCs on human umbilical vein endothelial cells (HUVECs) and normal human epithelial keratinocytes (NHEKs). (**A**) Tubular formation of HUVECs observed at 5 h after CM treatment. (**B**) Quantification of the tube formation in HUVEC compared to the 0% group (*n* = 3, * *p* < 0.05 compared to the 0% group, one-way ANOVA test). (**C**) Immunocytochemistry (ICC) of involucrin (green) in NHEK merged with DAPI (nucleus, blue) and phalloidin (F-actin, red). (**D**) Relative data of involucrin expression (* *p* < 0.05 compared to the 0% group, one-way ANOVA test). (**E**) Proliferation of NHEKs 3 d after CM treatment (*n* = 4, * *p* < 0.05 compared to the 0% group).

**Figure 7 pharmaceutics-13-01155-f007:**
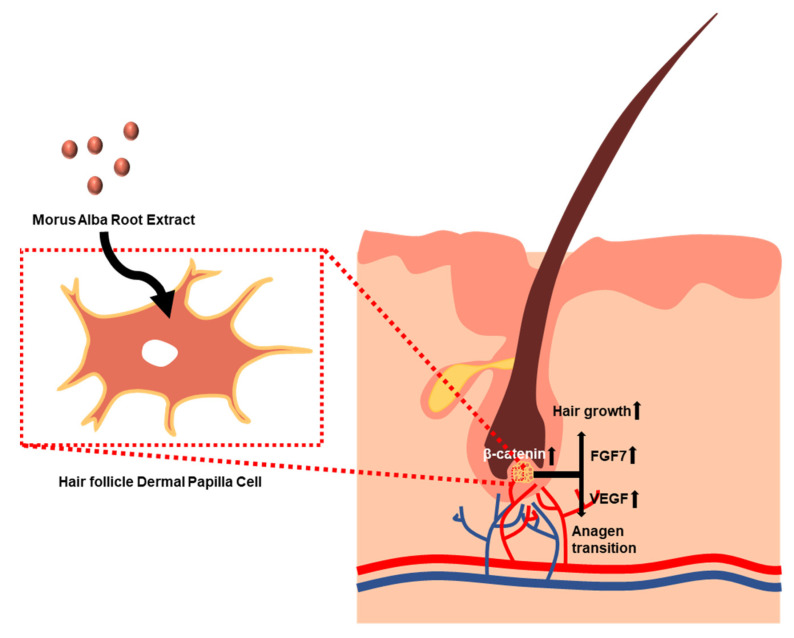
Illustration of the effects of MARE on the HFDPCs and hair growth-inducing potential.

**Table 1 pharmaceutics-13-01155-t001:** Primers used in quantitative reverse transcription polymerase chain reaction.

Gene	Forward Primer (5′ → 3′)	Reverse Primer (5′ → 3′)
*GAPDH*	5′-GTCGGAGTCAACGGATTTGG-3′	5′-GGGTGGAATCAATTGGAACAT-3′
*VEGF*	5′-GAGGGCAGAATCATCACGAAG T-3′	5′-CACCAGGGTCTCGATTGGAT-3′
*FGF7*	5′-CGCAAATGGATACTGACACG-3′	5′-GGGCTGGAACAGTTCACACT-3′
*HGF*	5′-GATGGCCAGCCGAGGC-3′	5′-TCAGCCCATGTTTTAATTGCA-3′
*FGF2*	5′-GACGGCAGAGTTGACGG-3′	5′-CTCTCTCTTCTGCTTGAAGTT-3′
*β-Catenin*	5′-TTGTGCGGCGCCATTTTAAG-3′	5′-TCCTCAGACCTTCCTCCGTC-3′

**Table 2 pharmaceutics-13-01155-t002:** Calibration curves and concentration of chlorogenic acid and umbelliferone.

Compounds	Regression Equation (*x*: Concentration, *y*: Peak Area)	Linear Range (ng/mL)	Correlation Coefficient (*R*^2^)	Dilution (mL/g)	Area of HPLC	Concentration (ng/mL)
Chlorogenic acid	*y* = 41,567.3320*x* + 625.4511	0.062–1.235	0.9991	200.0	8117	36.05
Umbelliferone	*y* = 28,646.2256*x* + 915.3077	0.032–0.322	0.9931	2.000	2380	0.10

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
