# Peer review of "Morus alba Root Extract Induces the Anagen Phase in the Human Hair Follicle Dermal Papilla Cells"

_pharmaceutics, 2021, doi:10.3390/pharmaceutics13081155_

Round 1
Reviewer 1 Report
The authors here detail the use of root extract Morus alba in inducing the transition of telogen- anagen phase within the human hair follicle dermal papilla cell. Indeed the restoration of hair follicles through the induction of the anagen phase would seem to be an extremely promising approach to the prevention of hair loss. Optimized concentrations of root extract were demonstrated by the authors to activate/enhance expression of growth factors crucial for induction of the telogen-anagen transition.
Overall the manuscript does provide a concise investigation into the potential therapeutic use of morus alba root extract for the prevention of hair loss. The following comments however would need to be addressed prior to publication.
As a general comment, the methods section does require improvements in its consistency and description. Numerous examples of established reagents such as
"radioimmunoprecipitation assay buffer"
"bicinchoninic acid assay"
"polyvinylidene fluoride"
can all be detailed as their acronyms RIPA, BCA, PVDF respectively.
Moreover in the description of western blotting, no detail is given as to where membranes were blocked with a blocking solution or what reagent was used for generation of luminescence.
Line 208-211: Authors need to remove the description of what should be detailed within the results section.
Figure 2,3. The overall quality of the images shown in the red filter for figure 2 and green filter for figure 3 are not suitable for publication. Image quality is not sufficient to make out any staining and needs to be rectified by the authors.
Figure 5 a,b. The authors need to add Y and X axis labels to figures respectively to adequately describe what the figure is detailing.
Author Response
Thank you for your time and consideration for revising our manuscript.
Please see the attachment.

Reviewer 2 Report
Morus alba is well known for its effect on hair regrowth and has been documented even in animal studies. This manuscript elucidates specific effects between cellular components of hair follicles, for more detailed mechanistic knowledge.
Its novelty and impact is somewhat limited, though it brings together concepts that were disperse throughout the literature.
The definition of objectives and conclusion should be more incisive on what the novelty is. For example, the final sentence in conclusions “This study suggests that MARE has great potential for promoting hair growth based on its cellular effects.” is too vague and could be drawn only from supporting literature.
The study overall identifies all important citations; missing from line 39 (when citing 12 and 13) is reference DOI: 10.1177/1087057112464574, which also has some information on VEGF and wnt signaling induced by minoxidil and herbal treatments.
There is no description regarding the quantification of cells from the images, for example figure 2.
Regarding cytotoxicity, ISO guidelines and general consensus identifies <80% viability as cytotoxic, and the results if expressed as % would be more easily compared with other studies. In figure 2c it is very difficult to observe cell morphology properly, and the obvious variations in cell density should also be mentioned to strengthen the discussion.
In figure 4, the axis units should be homogenous (they are all different); in figure 6 B, D and E should also be corrected in the same manner.
Author Response

(The authors gave the same response as above.)
